# *Tropheryma whipplei* Colonization in Adults and Children: A Prospective Study

**DOI:** 10.3390/microorganisms12071395

**Published:** 2024-07-09

**Authors:** Lucia Moro, Giorgio Zavarise, Giada Castagna, Elena Pomari, Francesca Perandin, Chiara Piubelli, Cristina Mazzi, Anna Beltrame

**Affiliations:** 1Department of Infectious, Tropical Diseases and Microbiology, I.R.C.C.S. Sacro Cuore Don Calabria Hospital, 37024 Negrar di Valpolicella, VR, Italy; elena.pomari@sacrocuore.it (E.P.); francesca.perandin@sacrocuore.it (F.P.); chiara.piubelli@sacrocuore.it (C.P.); cristina.mazzi@sacrocuore.it (C.M.); 2Pediatric Department, I.R.C.C.S. Sacro Cuore Don Calabria Hospital, 37024 Negrar di Valpolicella, VR, Italy; giorgio.zavarise@sacrocuore.it; 3Obstetrics Unit, Women’s and Children’s Health Department Gynecology, University of Padua, 35122 Padova, PD, Italy; giada.castagna.1@studenti.unipd.it; 4College of Public Health, University of South Florida, Tampa, FL 33620, USA; beltramea@usf.edu

**Keywords:** Whipple disease, carrier, colonization, molecular diagnosis, epidemiology, PCR

## Abstract

We conducted a prospective cohort study at the IRCCS Sacro Cuore Don Calabria Hospital in Negrar di Valpolicella from 2019 to 2021 to investigate the duration of *T. whipplei* colonization. In addition, the correlation between persistent colonization and the continent of origin, current treatment regimen, clinical manifestations, and parasite coinfection was evaluated. The cohort included subjects who were tested in a previous study (years 2014–2016) and found to be positive for *T. whipplei* DNA in fecal samples. Thirty-three subjects were enrolled in a prospective study between 2019 and 2021. Feces, saliva, urine, and blood were collected at baseline and after 12 months. Medical history, current treatment, and symptoms were recorded. Among them, 25% showed persistent intestinal or oral colonization, 50% had no colonization at both visits, and 25% had intermittent colonization. No association was found between persistent *T. whipplei* colonization and subjects’ continent of origin, current treatment regimen, initial clinical manifestations, and parasite coinfection. The longest duration of persistent *T. whipplei* intestinal colonization exceeded six years, with 11 subjects presenting persistent positivity for more than three years, including 1 minor. Our research was limited by the lack of a strain-specific identification of *T. whipplei* that made it impossible to distinguish between persistence of the same *T. whipplei* strain, reinfection from household exposure, or infection by a new strain. Larger prospective studies are needed to further explore the implications of this persistence and determine the key factors influencing the duration of colonization and its potential health impacts.

## 1. Introduction

*Tropheryma whipplei*, a rod-shaped Gram-positive bacterium, causes Whipple disease (WD), a rare infectious disease, characterized by gastrointestinal symptoms, such as diarrhea and significant weight loss along with joint pain and neurologic symptoms [1]. If left untreated, WD can be fatal [1]. In Europe and the United States, the estimated annual incidence of WD ranges from 1 to 6 cases per 10 million people [2]. However, the true incidence is likely to be underestimated because early-stage symptoms of WD are typically nonspecific [1], compounded by the challenges associated with microbiological diagnosis. The integration of molecular biology into patient screening has revolutionized diagnostic precision, uncovering a broad spectrum of clinical manifestations of WD [3]. Samples such as fecal matter, saliva, blood, and urine are increasingly recognized as valuable for comprehensive diagnostic assessments in WD [3,4].

The introduction of non-invasive microbiological assays has revealed the widespread presence of *T. whipplei* within the intestinal tract even among asymptomatic healthy individuals [5].

The colonization rate is influenced by different risk factors, including geographical location and age [5], as well as coinfections with pathogens that share transmission routes and risk factors, such as *Giardia duodenalis*, *Entamoeba histolytica*, and *H. pylori* [6,7,8].

In our previous study, we retrospectively assessed the prevalence of *T. whipplei* using fecal samples from 1240 patients visiting the hospital for various health reasons from 2014 to 2016, identifying an overall prevalence of 6.9% [6]. The prevalence was notably higher among children under 10 years (17.3%) and migrants (9.3%) [6]. However, comprehensive clinical information about the tested subjects was lacking.

Fenollar et al. studied a cohort of eight sewage workers previously identified as fecal carriers of *T. whipplei* for more than 1 year. They assessed the serological status against *T. whipplei* and genotyped the bacteria. Among these subjects, four remained asymptomatically colonized with the same strain, while two others cleared the infection. Additionally, two individuals were re-infected with a different strain of *T. whipplei* [9].

The transition from *T. whipplei* carriage to active disease poses central questions that remain unresolved, primarily due to the disease’s low incidence.

The primary aim of the present study was to investigate *T. whipplei* colonization over time in a cohort of both adults and children who resulted to be positive to *T. whipplei* in fecal samples in our previous study.

Moreover, we aimed to determine whether a possible persistent colonization could be associated with symptoms or correlated with specific patient characteristics, such as continent of origin, concomitant treatment (especially immunosuppressive treatment, including steroids), or coinfections with parasites.

## 2. Materials and Methods

### 2.1. Study Design and Setting of the Study

This prospective cohort study was conducted at the IRCCS Sacro Cuore Don Calabria Hospital in Negrar di Valpolicella, Verona, Italy from 2019 to 2021.

### 2.2. Ethics Statement

This study was conducted in accordance with the ethical principles of the Declaration of Helsinki. Ethical approval was secured from the “Comitato Etico Provinciale di Verona e Rovigo” under protocol Prot. Negrar 2019-36 dated 29 July 2019. Informed consent was obtained from all individuals enrolled in the study. For participants who were minors, consent was obtained from their parents or legal guardians.

### 2.3. Study Population, Data, and Samples Collection

The subjects eligible for enrollment in the study were individuals who had previously been identified as carrying *T. whipplei* in their fecal samples (n = 85, including 57 adults and 28 pediatrics) during a previous retrospective and cross-sectional study conducted by our group between January 2014 and April 2016 [6].

The eligible subjects were contacted by phone for enrollment in the prospective study between 2019 and 2021.

Exclusions encompassed patients declining participation and those with a prior diagnosis and treatment history of WD.

Subjects enrolled in the prospective study consulted with infectious disease specialists or pediatricians in a baseline visit to gather detailed medical histories, encompassing demographic details like age, gender, and nationality, along with information about comorbidities, history of parasitosis, and previous treatments (immunosuppressive and systemic steroid therapy). Participants were also asked to provide details about their current treatment and symptoms and to provide biological samples (feces, saliva, urine, and blood) both at the baseline visit and after a 12-month follow-up period.

To ensure that the biological samples were suitable for the analysis, participants were required to abstain from taking antibiotics or antiparasitic treatments for a month before each sample collection. If participants had taken these medications, they were asked to undergo a new evaluation after one month to provide appropriate samples. Patients with symptoms consistent with WD were offered invasive diagnostic procedures, in accordance with the established international recommendations [1], to ensure prompt detection and treatment of WD. Individuals who received a new WD diagnosis or underwent prolonged antibiotic treatment were subsequently excluded from the study. Data collection and management for the study were performed using the OpenClinica open-source software, version 3.1 (Copyright OpenClinica LLC and collaborators, Waltham, MA, USA, www.OpenClinica.com).

### 2.4. Laboratory Methods

#### 2.4.1. Sample Collection and DNA Extraction

According to the routine procedure of our laboratory, peripheral blood in EDTA, saliva, urine, and fecal matter was collected. For DNA extraction, 500 µL of blood, 200 µL of saliva, and 200 µL of urine were extracted without a pre-treatment, whilst 200 mg of fecal matter was suspended in 200 µL of phosphate-buffered saline containing 2% polyvinylpolypyrolidone (Sigma-Aldrich, St. Louis, MO, USA) and frozen overnight at −20 °C, followed by boiling for 10 min at 95 °C in heatblock and bead beating. In each sample, the internal control Phocine Herpes Virus type-1 (PhHV-1) was added before starting the extraction [10] in order to verify the good performance of the extraction and the presence of inhibitory molecules for the polymerase enzyme. The DNA was extracted by an automatic extractor (MagnaPure LC 2-Roche) using the DNA isolation kit I (Roche, Basel, Switzerland). The DNA was eluted in a final volume of 100 µL and stored at −20 °C for subsequent molecular tests.

#### 2.4.2. Real-Time PCR (rt-PCR)

Each DNA was amplified by real-time PCR. All amplification reactions were performed in 25 μL, with 5 μL of sample DNA. All primers/probes are summarized in Appendix A. Two specific rt-PCRs for *T. whipplei* [6] were performed in all the biological specimens, with the first PCR targeting the 105 bp sequence of the bacterium. The amplification conditions involved an initial denaturation step at 95 °C for 15 min, followed by 40 cycles of denaturation at 95 °C for 15 s and annealing and elongation at 60 °C for 60 s. If the result of the first PCR was positive, it was systematically confirmed by a second rt-PCR targeting a different DNA sequence. The same amplification conditions described above were used. Regarding the molecular diagnostic screening for intestinal parasites on the fecal sample, multiplex rt-PCRs were performed by adapting the reported protocols [10,11,12,13,14]. Rt-PCRs were performed using SsoAdvanced Universal Probes Supermix (Bio-Rad Laboratories, Milan, Italy) and the CFX 96 detection system (Bio-Rad Laboratories). Positive and negative controls were included in all the experiments. As a control for inhibitors and amplification, the exogenous PhHV-1 DNA was amplified with the appropriate primers/probe mix [10].

### 2.5. Definitions

Persistent colonization: subjects who tested positive for *T. whipplei* using PCR in their fecal or saliva samples both at baseline and follow-up visit.

Intermittent colonization: subjects who tested positive for *T. whipplei* using PCR in their fecal or saliva samples at either baseline or follow-up visit.

No colonization: subjects who tested negative for *T. whipplei* using PCR in their fecal and saliva samples at both baseline and follow-up visits.

### 2.6. Statistical Analyses

Statistical analyses were conducted using R software (version 4.2.3). Age was summarized as the mean and standard deviation (SD), and categorical variables were presented as absolute frequencies and percentages. Associations between persistent positive and intermittent or persistent negative qualitative variables were examined using Fisher’s exact test. Associations between baseline and follow-up were assessed using the McNemar test. Statistical significance was defined as a *p*-value < 0.05.

## 3. Results

### 3.1. Description of Study Population

Figure 1 illustrates the flowchart of participants, starting with the assessment of 85 subjects (57 adults, 28 pediatrics) colonized by *T. whipplei* from the 2014 to 2016 period [6]. Among these participants, 46 individuals (54.1%) did not respond to the phone call, and 5 (5.9%) declined to participate. Out of the initial cohort of 34 subjects (40%) meeting the study’s eligibility criteria, 1 individual was excluded due to a lack of fecal sample provision at baseline.

Table 1 presents a comprehensive overview of the demographic data, comorbidity profiles, and epidemiological information for the enrolled participants (n = 33, including 17 adults and 16 pediatrics). Additionally, Table 2 provides insights into the current treatment, symptoms, and microbiological results of the same participants at the baseline visit. Five participants were excluded from the follow-up analysis (Figure 1). Two participants were lost to follow-up, two were unable to provide the required fecal samples, and one participant initiated prolonged antibiotic treatment (Figure 1). Consequently, the current treatment, symptoms, and microbiological results of the remaining 28 subjects who underwent follow-up evaluation are detailed in Table 2. The baseline and follow-up visits were conducted, on average, 4.27 and 5.32 years, respectively, after the initial detection of intestinal colonization during the 2014–2016 study.

The mean age of the 33 participants was 29.6 years (SD 21.8): 51.5% adults and 48.5% individuals under 18 years of age. In terms of sex distribution, 54.5% were male. The participant demographics indicated that 51.5% of the cohort were Italian, 30.3% were born in Africa, 15.2% in Central and Southern America, and 3% were in Asia. At least one comorbidity was reported by 48.5% of the participants, with the most common conditions being asthma (9.1%), chronic hepatitis B (6.1%), and gastro-intestinal disturbances such as gastritis, duodenitis, or gastroesophageal reflux (9.1%). A history of parasitosis was reported by 18.2% of the participants, while immunosuppressive or steroid therapy histories were sporadically documented (5.9% and 8.8%, respectively).

At the baseline visit, current treatment was reported by 12.1% of patients (three undergoing systemic steroid therapy and one receiving immunosuppressive treatment) (Table 2). In the follow-up analysis, only one patient reported immunosuppressive therapy. During the baseline visit, 33.4% of the participants reported experiencing symptoms, and this percentage slightly increased to 39.3% during the follow-up visit. The most commonly reported symptom was abdominal pain (27.3% at baseline, 14.3% at follow-up), followed by diarrheal syndrome (15.2% at baseline, 14.3% at follow-up) and arthralgia (15.2% at baseline, 14.3% at follow-up). The duration and temporal distribution of abdominal pain and diarrhea varied among the participants at baseline and in the follow-up analysis (Table 2). Additional symptoms identified at baseline included the following: weight loss (12.1%), myoclonus (6.1%), arthritis (3.0%), cognitive decline (3.0%), eye disorders (3.0%), and abnormal ocular movements (3.0%). Psychiatric disorders were reported by 9.1% of the individuals at baseline, and by 3.6% at the follow-up. Notably, no cases of fever lasting longer than one week were reported.

### 3.2. PCR Analysis for T. whipplei

Out of the 33 subjects enrolled in the study, 14 (42.4%) tested positive for *T. whipplei* using PCR on their fecal samples at baseline. At the follow-up, among the remaining 28 subjects, 9 (32.1%) tested positive. Five subjects tested positive at baseline but negative at follow-up, while three patients tested negative at baseline but were positive at follow-up. Additionally, three subjects who tested positive for *T. whipplei* at baseline did not complete the follow-up visit. Figure 2 illustrates the temporal fluctuations of the *T. whipplei* PCR results in fecal samples from 33 subjects over the period of 2014–2016 (initial positive molecular detection) to 2020–2021 (baseline and follow-up of the current study). The figure highlights the dynamic nature of PCR outcomes, illustrating both persistence and changes in colonization status. Remarkably, the longest observed period of persistent *T. whipplei* intestinal colonization, from the initial positive PCR, spanned over 6 years, precisely 6.17 years. A subset of 11 patients showed persistent positive results for more than 3 years, including one minor who had persistent intestinal *T. whipplei* colonization for 5.9 years.

Regarding the saliva specimens, initial examination detected *T. whipplei* DNA in five (15.2%) of the participants. This number decreased to two (7.1%) by the six-month follow-up. Only one subject (3.6%) exhibited the persistent presence of *T. whipplei* DNA in both fecal and saliva samples from the initial visit through to the follow-up. Remarkably, this individual did not exhibit any symptoms throughout the entire study period.

Another subject tested positive for *T. whipplei* in both fecal and saliva samples at baseline. Due to articular and gastrointestinal symptoms initially suggestive of WD, this subject underwent invasive diagnostic procedures, started antibiotic treatment, and was subsequently withdrawn from the study. Ultimately, WD was ruled out, and antibiotic treatment was discontinued.

Additionally, another subject tested positive for *T. whipplei* in both fecal and saliva samples only at the follow-up visit but did not report any symptoms.

During the study period, 7 out of 28 subjects (25%) showed persistent intestinal or oral colonization by *T. whipplei*. No colonization was found in 14 subjects (50%), while the remaining participants displayed intermittent colonization (25%).

Blood and urine samples consistently tested negative for *T. whipplei* PCR at both baseline and follow-up stages of the study.

### 3.3. PCR Analysis for Parasites in the Fecal Samples

Twelve (36.4%) participants tested positive for one or more protozoa in their fecal samples at baseline: four for *Blastocytis* spp.; four for *D. fragilis*; and four for more than one parasite (two subjects were coinfected with *D. fragilis* and *Blastocystis* spp., one with *D. fragilis* and *H. nana*, and one with *Blastocystis* spp. and *E. dispar*).

### 3.4. Association Analysis

The analysis performed with Fisher’s exact test revealed no significant association between the persistent colonization of *T. whipplei* and the subjects’ continent of origin, current treatment regimen (including systemic steroid and immunosuppressive therapy), initial clinical manifestations, or parasite coinfection (Table 3).

## 4. Discussion

To the best of our knowledge, this study represents the only comprehensive prospective investigation of *T. whipplei* colonization over time and across different biological samples in both adults and children.

The rate of intestinal colonization by *T. whipplei* has been shown to be influenced by environmental factors and patient age [6,11,12,13,14,15,16]. For instance, intestinal colonization prevalence is higher in environments with poor sanitation, such as 17.4% in adults living in rural areas of Senegal compared to 4% in the French general adult population [12,16]. It is also more common in relatives of WD patients or *T. whipplei* chronic carriers [16,17] and in children compared to adults, with a prevalence of 12.7% in children under 10 years of age compared with 5.9% in older children [6]. Additionally, an higher prevalence of *T. whipplei* intestinal colonization has been reported in migrants from low- and middle-income countries with a rate of 9.3% in migrants compared with 4.9% in the Italian population [6], and 27% in migrant children over 12 years old in Greece [18]. However, information regarding the duration of colonization and its clinical implications remains inadequately documented.

In our study, approximately 42% of the subjects initially colonized by *T. whipplei* still tested positive for the bacteria in their fecal samples more than four years after the first assessment. This proportion decreased to 32% by the follow-up visit one year later without any treatment. The longest duration of persistent *T. whipplei* intestinal colonization recorded was over six years. Persistent colonization, defined as the presence of *T. whipplei* in both the baseline and follow-up samples, was found in about 25% of the 28 subjects who returned for the follow-up. This gradual reduction in colonization rates is consistent with the findings of Fenollar et al., who investigated *T. whipplei* carriage in sewage workers [9]. However, their study was limited to a selected group of adults and they followed only eight individuals with intestinal colonization for more than one year [9].

Moreover, in our study, we observed persistent positivity for *T. whipplei* in only one minor through fecal samples analysis. This patient was 13 years old at the time of the initial detection of intestinal colonization and remained positive until the follow-up visit, which occurred 5.9 years later. To our knowledge, there are no available literature data regarding the duration of *T. whipplei* colonization in children.

Our study found no association between persistent or intermittent colonization and specific continents of origin or concurrent immunosuppressive treatments, although the limited sample size may influence these findings. Additionally, we did not identify a significant association between *T. whipplei* carriage and symptoms.

At baseline, five individuals (two adults, three minors) tested positive for *T. whipplei* in saliva samples, with two of them also testing positive in fecal and saliva samples. However, at follow-up, only two individuals (one adult, one minor) remained positive in saliva samples. Despite the small sample size, this result is interesting as it appears to confirm a gradual decline over time in the rate of oral colonization, mirroring the pattern observed for intestinal colonization. Notably, only one adult maintained persistent double *T. whipplei* positivity (in both fecal and saliva samples) without developing WD. This finding aligns with observations by Fenollar et al. [3], who noted that the asymptomatic carriage of *T. whipplei* DNA in saliva and fecal samples is rare in the general population.

The relationship between *T. whipplei* colonization and WD remains unclear. *T. whipplei* is a widespread bacterium, whereas WD is a very rare disease [1]. Interestingly, the rate of *T. whipplei* colonization is higher in countries where the prevalence of WD is lower [1,2].

The current hypothesis regarding the pathogenesis of WD suggests that many individuals are exposed to *T. whipplei* in early childhood, but only a small fraction, possibly due to genetic predisposition, will develop the disease [1]. It remains uncertain whether a prolonged duration of the colonization indicates an increased risk of developing WD or if it plays a causative role in its onset. Similarly, it is unclear whether early-life exposure to *T. whipplei*, common in regions with poor hygiene standards, could potentially confer protection against the development of the disease [1]. In our study, none of the patients who tested positive for *T. whipplei* in fecal, saliva, or both samples developed WD. The average follow-up period from the first positive fecal sample was 5 years. Additionally, none of our patients exhibited *T. whipplei* positivity in blood or urine, which typically correlates with active WD [1]. These findings align with those reported by Fenollar et al., who observed that none of the sewage workers carrying intestinal *T. whipplei* developed symptoms indicative of WD [9]. However, clinical monitoring of individuals enrolled in our study is ongoing, and no cases of WD have been identified to date.

The primary limitation of our study lies in its design, which was constrained by the necessity to follow a rare disease that develops over a prolonged period of time. This difficulty could be overcome by conducting a multi-centric study with an extensive follow-up.

Moreover, our research was limited by the absence of the strain-specific identification of *T. whipplei* at baseline and follow-up time points, making it impossible to distinguish between the persistence of the same *T. whipplei* strain, reinfection from household exposure, or infection by a new strain. The use of a Next-Generation Sequencing (NGS) approach in future studies could overcome this limitation.

Moreover, the selective patient demographic poses challenges to generalize our findings to a broader population.

Unresolved critical issues include understanding why only some carriers of *T. whipplei* develop the disease and the temporal dynamics that govern this progression in adults and children. Future long-term cohort studies with the strain-specific identification of *T. whipplei* are necessary to elucidate the factors contributing to the transition from asymptomatic colonization to active disease. Despite their inherent challenges, these studies are essential, as they hold the potential to significantly enhance our understanding and management of *T. whipplei* infections. Furthermore, the absence of established guidelines for managing asymptomatic carriers underscores the urgent need to develop comprehensive public health protocols. These protocols should focus on regular monitoring, early detection of symptoms, and preventive measures to mitigate the risk of progression to active disease in high-risk patients. By addressing these gaps, we can ensure a better management of individuals colonized by *T. whipplei* and improve the diagnosis of WD.

## Figures and Tables

**Figure 1 microorganisms-12-01395-f001:**
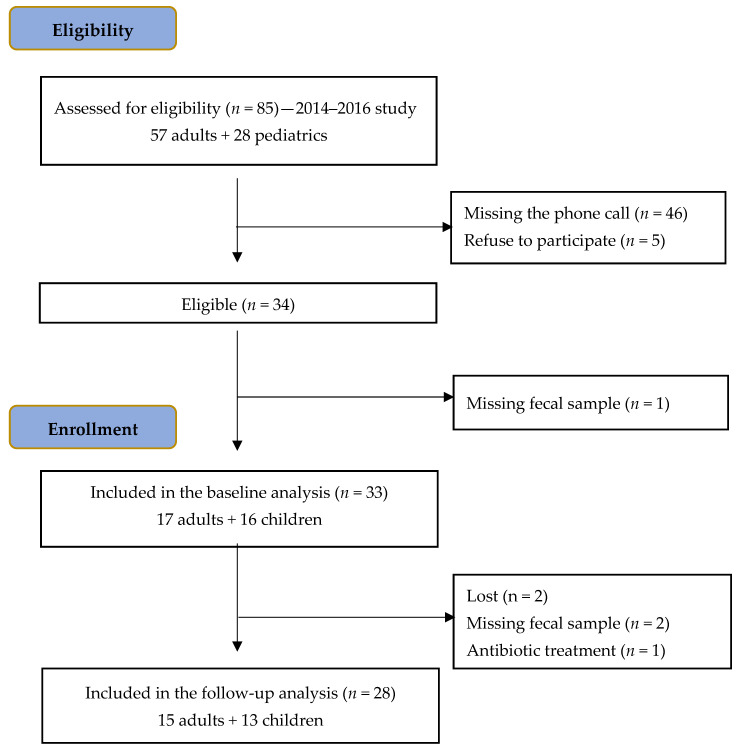
Flowchart of participants.

**Figure 2 microorganisms-12-01395-f002:**
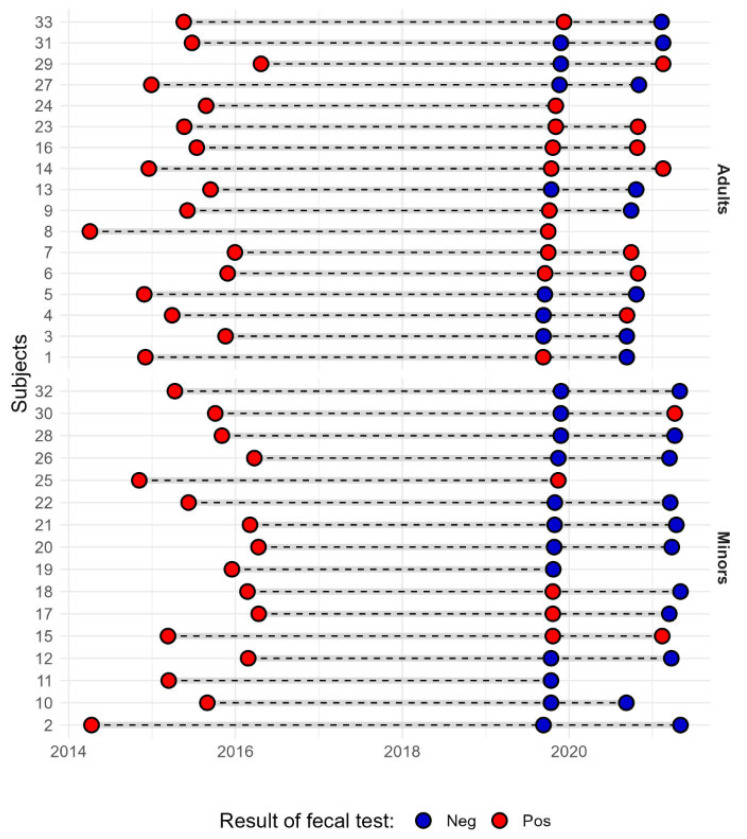
Dynamics of *T. whipplei* PCR results in 33 fecal samples of adults (n = 17) and children (n = 16) from 2014 to 2020 period. Red dots indicated positive PCR results for *T. whipplei*, while blue dots denote negative results.

**Table 1 microorganisms-12-01395-t001:** Demographic data, comorbidity, and epidemiological information of the overall patients at the baseline.

Variables	Overall Sample	
	n	%
Overall Participants	33	100
Age, mean (SD)	29.6 (21.8)	-
Age group (years)		
<18	16	48.5
≥18	17	51.5
Male	18	54.5
Nationality		
Italy	17	51.5
Africa	10	30.3
Central and Southern America	5	15.2
Asia	1	3
Comorbidity	16	48.5
Asthma	3	
Chronic hepatitis B	2	
Gastritis, duodenitis, or GER	3	
Hypertension	2	
Parasitosis	3	
Psoriatic arthritis	1	
Type II diabetes mellitus	1	
Hypothyroidism	1	
Monoclonal lymphocytosis B	1	
Not specific	1	
History of intestinal parasitosis	6	18.2
History of immunosuppressive therapy	2	5.9
History of systemic steroid	3	8.8
History of tropical travel	17	51.5

SD, standard deviation; GER, gastroesophageal reflux.

**Table 2 microorganisms-12-01395-t002:** Clinical and microbiological characteristics of the overall patients at the baseline and after 12 months follow-up. Associations were tested using the McNemar test.

Variables	Baseline		Follow-Up at 12 Months		*p*-Value
	n	%	n	%	
Overall participants	33	100	28	100	
Current treatment	4	12.1	1	3.6	1.000
Systemic steroid	3		1		
Immunosuppressive therapy	1		1		
Symptoms or signs	11	33.4	11	39.3	1.000
Abdominal pain	9	27.3	4	14.3	0.343
Abdominal pain duration *					
1 days	5		1		
2–15 days	2		0		
>30 days	2		0		
NA	0		3		
Abdominal pain temporal distribution *					
Single episode	2		0		
2–4 episodes	1		1		
≥5 episodes	6		1		
NA	0		2		
Diarrheal syndrome	5	15.2	4	14.3	1.000
Diarrheal syndrome duration					
<15 days	4		2		
15–30 days	0		0		
>30 days	1		0		
NA	0		2		
Arthralgias	5	15.2	4	14.3	1.000
Weight loss	4	12.1	0		-
Psychiatric disorders	3	9.0	1	3.6	1.000
Myoclonus	2	6.0	0		-
Fever longer than 1 week	0		0		-
Arthritis	1	3.0	0		-
Cognitive decline	1	3.0	0		-
Eye disorders	1	3.0	1	3.6	1.000
Abnormal ocular movements	1	3.0	0		-
Microbiological results					
Fecal *T. whipplei* PCR-positive	14	42.4	9	32.1	0.724
Saliva *T. whipplei* PCR-positive	5	15.2	2	7.1	1.000
Blood *T. whipplei* PCR-positive	0		0		-
Urine *T. whipplei* PCR-positive	0		0		-

* one or more patients did not answer. NA, not applicable.

**Table 3 microorganisms-12-01395-t003:** Fisher’s exact tests of association between PCR *T. whipplei* persistence in fecal and saliva samples and continent of origin, treatment, symptoms, and coinfection of 28 subjects evaluated at the baseline and at the follow-up.

Variables	PCR *T. whipplei*/ Persistent Positive		PCR *T. whipplei* Intermittent or Persistent Negative		*p*-Value
	n	%	n	%	
Total participants	7	25	21	75	
Continent of origin					0.254
South/Central America	1	14.3	4	19.0	
Africa	4	57.1	5	23.8	
Europe (Italy)	2	28.6	12	57.1	
Current treatment					
Systemic steroid	0		2	9.5	1
Immunosuppressive therapy	0		1	4.8	1
Symptoms or signs	2	28.6	8	38.1	1
Abdominal pain	2	28.6	6	28.6	1
Arthralgias	0		4	19.0	0.550
Weight loss	1	14.3	2	9.5	1
Psychiatric disorders	1	14.3	1	4.8	1
Myoclonus	0		1	4.8	1
Parasite coinfection at baseline	1	14.3	10	47.6	0.190

## Data Availability

The dataset used in this study is available upon request to the author.

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
