# Peer review of "Tropheryma whipplei Colonization in Adults and Children: A Prospective Study"

_microorganisms, 2024, doi:10.3390/microorganisms12071395_

Round 1

Reviewer 1 Report

Comments and Suggestions for Authors

Moro et al. have reported on – likely – persistence of T. whipplei colonization, a topic on which existing data is still limited. My comments are as follows:

1.) I leave it to the editor’s discretion whether the limited provided datasets really justifies a full-length paper format or whether a “communication” format might be more appropriate. Personally, I recommend for the latter.

2.) Basically, the authors stress the important limitations of their work in the last-but-one paragraph of their discussion, which is small sample size and lacking differentiation of persistence versus re-infection. They should mention that the first limitation can be overcome by multi-centric study-design and the last limitation by the inclusion of NGS-based diagnostic approaches in future follow-up assessments.

3.) As many readers just focus in the abstract, the authors should include the lacking discrimination of persistence in re-infection in the abstract as well in order to avoid over-interpretation of the results.

4.) Although I am not a native speaker myself, I feel that thorough language proof-reading either by the authors or by the journal would increase the readability of the work.

Author Response

Reviewer's comments: Moro et al. have reported on – likely – persistence of T. whipplei colonization, a topic on which existing data is still limited.

Authors' answer: We sincerely thank the reviewer for the constructive comments. We believe they have significantly helped us to improve the quality of the manuscript.

Reviewer's comments:
My comments are as follows:

1.) I leave it to the editor’s discretion whether the limited provided datasets really justifies a full-length paper format or whether a “communication” format might be more appropriate. Personally, I recommend for the latter.

Authors' answer: We understand the reviewer's point of view however, we prefer to present this work as a full-length article rather than a short communication because we believe that this format allows us to better illustrate and clarify the microbiological results and their correlation with the clinical data.

Reviewer's comments 2.) Basically, the authors stress the important limitations of their work in the last-but-one paragraph of their discussion, which is small sample size and lacking differentiation of persistence versus re-infection. They should mention that the first limitation can be overcome by multi-centric study-design and the last limitation by the inclusion of NGS-based diagnostic approaches in future follow-up assessments.

Authors' answer: We followed the reviewer’s suggestion and included in the final discussion paragraph the possibility to overcome the indicated limitations in future multicentric studies and employing NGS to determine the specific T. whipplei strain.

Reviewer's comments 3.) As many readers just focus in the abstract, the authors should include the lacking discrimination of persistence in re-infection in the abstract as well in order to avoid over-interpretation of the results.

Authors' answer: We agree with reviewer’s comment. In the abstract, we included a sentence indicating that our results do not discriminate between a persistence of the same T. whipplei strain, a reinfection from household exposure, or an infection by a new strain.  

Reviewer's comments 4.) Although I am not a native speaker myself, I feel that thorough language proof-reading either by the authors or by the journal would increase the readability of the work.

Authors' answer: We have completed a thorough proof-reading of the full text to improve the English language.

Reviewer 2 Report

Comments and Suggestions for Authors

Dear authors, I share my observations from the document entitled Tropheryma whipplei colonization in adults and children: A prospective study:

In the document I could not verify what the objective was, this did not allow me to verify the conclusion.

In the methodological part, the Study population, data, and samples collection section is very confusing, the authors have to restructure this section.

The authors indicate a statistical methodology for data analysis, however the results are analyzed by descriptive statistics and they could not determine an association. The statistical part is very important in scientific publication, which is why the authors have to look for a methodology that can be applied, this would give more certainty to their results and the discussion of them.

Comments on the Quality of English Language

Moderate editing of English language required

Author Response

Dear authors, I share my observations from the document entitled Tropheryma whipplei colonization in adults and children: A prospective study:

Authors' answer: We wish to express our sincere gratitude to the reviewer for the constructive observations that have helped us to identify problematic parts of the manuscript and improve them. 

Reviewer's comments 1.) In the document I could not verify what the objective was, this did not allow me to verify the conclusion. In the methodological part, the Study population, data, and samples collection section is very confusing, the authors have to restructure this section.

Authors' answer: We carefully revised the introduction, the methods section and the abstract, in order to make the objectives, the study population and sample collection more clear and readable. We hope to have now improved the paper presentation.

Reviewer's comments 2.) The authors indicate a statistical methodology for data analysis, however the results are analyzed by descriptive statistics and they could not determine an association. The statistical part is very important in scientific publication, which is why the authors have to look for a methodology that can be applied, this would give more certainty to their results and the discussion of them.

Authors' answer: In terms of statistical analysis, p-values for Fisher exact test comparisons between persistent positive and intermittent or persistent negative were presented in Table 3 and commented on in Section 3.4. To make the use of statistical tests clearer, we have highlighted their use in the results section. Comparisons between baseline and follow-up characteristics have been added to Table 2 using a test for paired data. We also updated paragraph 2.6 with details about the performed statistical analyses.

Additional Reviewer's comments: Moderate editing of English language required
Authors' answer: We have completed a thorough proof-reading of the full text to improve the English language.